# Selenium and Vitamin E Concentrations in Miranda Jennies and Foals (*Equus asinus*) in Northeast Portugal

**DOI:** 10.3390/ani11061772

**Published:** 2021-06-14

**Authors:** Miguel Quaresma, Carmen Marín, Daniel Bacellar, Miguel Nóvoa, Francisco Javier Navas, Amy McLean

**Affiliations:** 1Animal and Veterinary Research Center (CECAV), Universidade de Trás-os-Montes e Alto Douro, 5000-801 Vila Real, Portugal; miguelq@utad.pt; 2AEPGA-Association for the Study and Protection of Donkeys, Atenor, 5225-011 Miranda do Douro, Portugal; danielbacellar3@gmail.com (D.B.); miguelnovoa@aepga.pt (M.N.); 3Genetics Department, Veterinary Sciences, Rabanales University Campus, University of Córdoba, Madrid-Cádiz Km. 396, 14014 Cordoba, Spain; carmen95_mn@hotmail.com; 4The Worldwide Donkey Breeds Project, Rabanales University Campus, University of Córdoba, Madrid-Cádiz Km. 396, 14014 Cordoba, Spain; acmclean@ucdavis.edu; 5Instituto de Investigación y Formación Agraria y Pesquera (IFAPA), Alameda del Obispo, 14004 Cordoba, Spain; 6Department of Animal Science, University of California Davis, Davis, CA 95617, USA

**Keywords:** foals, jennies, donkey health, pregnancy, lactation, selenium, white muscle disease, milk, vitamin E, donkeys

## Abstract

**Simple Summary:**

Despite the importance of donkeys through history and their productive resuscitation during the last decades, reference values for common elements are not yet readily available. Such a challenge becomes even more noticeable when practitioners aim at evaluating the physiological and pathological concentrations of certain elements across the different stages that a donkey can go through along its life. The aims of this study are to determine baseline selenium and vitamin E concentrations for Miranda donkeys both jennies and foals. Miranda donkeys are considered to be endangered and it is possible that selenium and vitamin E may be associated with foal survival. Critical points may be identified related to overdosing or deficient levels of selenium and vitamin E, at different stages of development of gestation in utero during fetal development, parturition, and post foaling. Our study suggests that vitamin E and Se levels can have a major impact and effect on foal health and mortality levels. Multiple factors including location, diet, management practices, parity and time of breeding, and age of jenny may affect blood profiles in jennies, which ultimately may affect the profiles of her foals.

**Abstract:**

The increase in donkeys treated by practitioners in recent years has led to an increased interest in finding more information on basic biochemical preliminary reference values. The aims of this study were to measure Se and Vit E levels in plasma from Miranda jennies peripartum and postpartum and in their foals to compare blood profiles of the jenny and foal related to the overall foal’s health. Twenty-two healthy peripartum and postpartum Miranda donkeys were sampled (12 jennies and 10 foals) in the northeast of Portugal (Atenor and Paradela) from May to November, 2018. Amounts of selenium in soil were significantly correlated (0.97) to concentrations of selenium in jennies (42.412 μg/L in Atenor and 9.612 μg/L in Paradela) and foals (19.378 μg/L in Atenor and 6.430 μg/L in Paradela). Selenium levels were lower in foals than adults and in males than females. Vitamin E was associated with overall foal health. Foals with a mean vitamin E of 3.585–5.307 mg/L showed signs of weakness, but carpal flexural deformities were observed when the average vitamin E was 11.520 mg/L. Low vitamin E levels (5.307 mg/L) in jennies were related to foal mortality. Diets, location, parity, and age affect blood profiles of jennies and, ultimately, foal health.

## 1. Introduction

Recent efforts have examined risks factors that may be associated with the high foal mortality rates that are being seen by Miranda donkey owners [1,2]. Great efforts to preserve endangered breeds of donkeys such as the Miranda has led to more scientific literature and specie-specific information on donkey health becoming available along with interest in donkeys being used and valued as production and performance animals [3]. Today, we find donkeys being kept and managed more intensively and carefully than ever before [4]. Unfortunately, there is still a lack of information related to the basic understanding of donkey health parameters [4]. Hence, even routine diagnostic tests such as blood chemistry and hematological test parameters may be based on laboratory standards set for horses, which complicates routine internal medicine and general health care.

Although, more studies have begun focusing on species specific information, still only a few have assessed vitamin and mineral profiles [5,6]. Selenium (Se) an vitamin E (Vit E) may play a essential role for metabolic pathway and for protection against oxidative stress, which may be promoted under overwork situations [7]. Portugal is typically selenium deficient in soil and forages [8]. Selenium is a functional trace mineral component of intracellular enzyme glutathione peroxidase (GPx). Vit E, a lipid-soluble vitamin, is one of the most important antioxidants and free-radical reducers found in cell membranes. It donates reducing equivalents to lipid peroxyl radicals, converting them to less toxic lipid hydroperoxides, thus protecting unsaturated membrane lipids against oxidative damage [9]. Se and Vit E antioxidant function prevents cell membranes from damage due to free radicals [10]. Selenium-vitamin E injections can be used for short-term supplementation [11]. Consequently, Vit E and Se work synergistically to reduce oxidative damage [12]. 

Vitamin E and Se are essential nutrients for equine [13]. Levels of both the nutrients in foals are related to foal-dam prepartum and postpartum conditions and factors which have often been addressed in literature. Age and sex determine the levels of Vit E and Se levels in foals that have lower levels of Vit E compared to adults [6]. Additionally, Se status of the fetus and neonate foals is affected at various stages of foal development from *in utero* to the point of ingesting both colostrum and milk [14,15]. When appropriate levels of Se and Vit E intake occur in gestating dams, conditions such as White Muscle Disease (WMD) and other motor neuron diseases in neonate animals (e.g., calves) may be prevented [16,17,18]. Defining appropriate or normal levels of Se and Vit E in donkeys is still being determined by sampling populations and measuring different parameters that may alter blood samples levels. Route of transmission vary for Se and Vit E in foals. Foal selenium uptake occurs via the placenta during gestation and Vit E intake occurs after birth during ingestion of the colostrum. Hence, Vit E and Se intake and absorption greatly depends upon Se and Vit E status of the dam along with diet or supplementation of Vit E specially in the form of alpha tocopherol [18,19,20]. Deficiencies may especially compromise the health of new born donkey foals, young growing donkeys, or lactating jennies. Thus, both are key to foal survival and are required for proper immunity to proper muscle and neuron function [21]. 

Clinical manifestations of decreased antioxidant activity associated with Se and Vit E deficiency cause neurological and impaired muscular symptoms and a significant decrease in immune response in horses [21,22]. White muscle disease can affect both horse [23] and donkey foals [24]. In some cases, weak muscles in the throat of the foal may lead to milk entering the trachea and then in the lungs causing aspiration pneumonia. Other conditions in horses include equine motor neuron disease (EMND) and degenerative myeloencephalopathy (EDM) [16,18,25,26]. Deficiencies in Se and Vit E have also been associated with significant decreases in immune response in non-supplemented horses [21,27].

Despite the lack of specific knowledge on the risk factors for Se and Vit E deficiency or overdose in donkeys, authors such as Shawaf [28] and Bazzano et al. [6] have reported interspecies and sex differences. Other authors have addressed additional factors such as the low levels of Se in the soil of certain regions to be sufficient to predispose newborn foals to develop pathological conditions, along with the possibility of donkeys having a lower Se blood profile when compared to horses [6]. 

The first aim of the study was to determine blood profiles of Se and Vit E in jennies and their foals and compare them. Second objective looked at establishing a relationship between Se and Vit E levels in jennies and their foals in relation to the mineral composition of the soil. The last objective compared the jennies’ influence on foal health/survivability by examining her parturition history, foaling record, and blood profile of Se or Vit E related to their foal’s health.

## 2. Materials and Methods

### 2.1. Animals

Blood samples were collected from 12 Miranda jennies and 10 foals. Two foals were not sampled due to premature deaths. Animals were housed with private owners throughout the Portuguese villages of Atenor and Paradela. Animals were sampled at their housing facilities. 

A 10 mL sample of blood was collected from the jugular vein of each jenny and/or foal into an EDTA (ethylenediaminetetraacetic acid) tube (VACUETTE; Greiner Bio-One GmbH, Kremsmunster, Austria). All blood samples were stored in a dark box refrigerated at 18 °C immediately after collection and then transported from sampling location to laboratory. The jennies (*n*= 12) ranged in age from 3 to 24 years (7.9 ± 5.9 years), with an average BCS of 2.9 ± 0.59 (1 to 5 scale). The foals (*n* = 10, *n* = 6 males and *n* = 4 females) were sampled between day one and three after birth, 2 ± 1 days, and BCS was found to be 2.3 ± 0.35 (1 to 5 scale).

The first samples were collected from the jennies at 35.4 ± 15.6 days before foaling, and the second sample was collected at the same time when the foal sample was collected, i.e., 43.2 ± 15.2 h after foaling. After birth, the foals were allowed to nurse, and then, they were separated for 3 h from their dams before sampling. This sampling was performed in the morning, between 9:00 a.m. and 10:00 a.m., before feeding. No animal was transported to perform sampling as all animals were sampled in their normal location.

The donkeys had daily access to pasture, supplemental forage, grain (e.g., oats), and salt blocks depending on the owner’s animal feeding habits. Information pertaining to diets and village where the donkeys came from was recorded in the owner survey. The types of diets observed were: oat (straw with grain), oat (straw without grain), and hay supplementation using salt blocks ad libitum or extra feed. Information in regards to potential risk factors were recorded. Parameters considered for the jenny and foal were breeding date, date of foaling, parity, foal sex, previous foal deaths, and current foal status such as weak, vigor, alive/dead, twins, and carpal flexure deformities.

Animal handling was performed in compliance with the national regulations and the European Council Guidelines (Directive 2010/63/EU) for protection of animals used for scientific purposes, and respecting Animal Care and Welfare protocols. No Ethic committee approval was sought, as sampling was performed as part of the routine health check-up of the females and donkey foals.

### 2.2. Laboratory Analysis

Once samples arrived at the laboratory, each sample was removed from the dark box and centrifuged for 10 min at 1500 g and 4 °C within 30 min from collection. For each animal, 1.5 mL plasma aliquot was stored at −80 °C until analysis [13]. Vitamin E assays were performed using high-performance liquid chromatography (HPLC) (VWR Hitachi Chromaster, Tokyo, Japan). For each sample, 200 μL of plasma was added with 400 μL of precipitating agent and centrifuged at 13,000× *g* for 10 min before HPLC analysis [13]. In the analytical session, a standardized calibrator (ClinCal Plasma Calibrator lyophilized for vitamin A and E; RECIPE, Munich, Germany) and a certified control (ClinChek Plasma Control lyophilized for vitamins; RECIPE, Munich, Germany) were used for quantitative determination and to verify the correctness of the calculation, respectively.

Plasma samples were diluted in 1:3 ratio with a solution containing double-distilled water and Triton-X 0.05% (Fluka Chemika; Buchs, Switzerland). The calibration curve was prepared starting from the Se standard for AAS Fluka (Sigma-Aldrich, Darmstadt, Germany). Four points of calibration were obtained, i.e., 0, 50, 100, and 200 μg/L. The quantification of plasma concentrations of Se was performed in atomic absorption (Zeeman effect, AA240Z; Agilent Technology, Santa Clara, CA, USA) using partitioned tubes pyrolytically coated with GTA (Agilent Technology, Santa Clara, CA, USA). The sensibility of the analysis was improved by means of Se ultrAAhc lamp-Se coded (Agilent Technology, Santa Clara, CA, USA). The inert gas required was argon. To reduce or eliminate volatilization or interference in the vapor phase, palladium was used as modifier (Palladium matrix modifier, Fluka; Sigma-Aldrich, Darmstadt, Germany) diluted 1:20 with double-distilled water.

### 2.3. Environmental (Soil Se Levels)

Environmental Se levels (soil) were determined using the protocols of the Technological and Nuclear Institute of the University of Lisbon (Portugal) as described in Ventura et al. [29]. Data base for the levels of selenium during the time of the study were consulted in the European Soil Data Centre (ESDAC).

### 2.4. Statistical Analysis

Due to the difficulties of working in location with an endangered breed with a small population scattered through different areas [1,2], the small sample size and sample distribution violated the parametric assumptions (Shapiro-Francia *p* < 0.05 and Levene’s test *p* < 0.05), therefore, Bayesian inference Pearson correlation function was used to characterize the posterior distribution of the linear correlation between environmental level (soil Se levels) and jennies’ and their foals’ concentrations of Se (μg/L) and Vit E (mg/L). 

Additionally, the correlation between jennies and their foals’ concentrations of Se (μg/L) and Vit E (mg/L) and age was also tested using the same Bayesian methods described above. Correlation coefficients were analyzed to determine whether a linear relationship should be expected among these parameters. Bayesian inference for Pearson correlation was performed using the Pearson correlation task from the Bayesian statistics procedure in SPSS Statistics, Version 25.0, (IBM Corporation, Armonk, NY, USA) [30]. A full description of the algorithms used by SPSS to perform Bayesian Inference on Pearson correlation in this study can be found in the public document IBM SPSS Statistics Algorithms version 25.0 by IBM Corp. [31].

Afterwards, Bayesian inference for ANOVA was run to test for statistical differences in the mean for Vit E and Se concentrations across the factors such as foal sex, BCS, age, forage type supply, diet supplementation, previous foal status, number of previous parturitions, foal death or foal status afterbirth, sampling location, age group, sampling month, breeding month, parturition month, and owner.

As suggested in document IBM SPSS Statistics Algorithms version 25.0 by IBM Corp. [31], Bayesian inference of ANOVA is approached as a special case of the general multiple linear regression model. A full description of the algorithms used by SPSS to perform Bayesian Inference on Analysis of Variance (ANOVA) in this study can be found in the public document IBM SPSS Statistics Algorithms version 25.0 by IBM Corp. [31]. The tolerance value for the numerical methods and the number of method iterations were set as a default by SPSS v25.0 [30]. 

The estimated effect of the factors considered in the predictive models, its interval, and the posterior distribution statistics were interpreted. The 95% credibility interval shows that there is a 95% probability that these regression coefficients (posterior distribution mean value for each factor) in the population lie within the corresponding intervals. When 0 is not contained in the credibility interval, a significant effect for such factor is detected. Bayes factor (BF) was computed to determine the likelihood of null and alternative hypotheses or one model versus another based on the prior distribution and the data. 

Bayesian estimation methods have been reported to require a much smaller ratio of parameters to observations (1:3 instead of 1:5), as they maximize the ability to determine significant effects for relatively limited sample sizes. These sample limitations are reflected in the broadening of confidence intervals, which must be accompanied by an acceptable Bayes factor value as suggested by Hox et al. [32] and Lee and Song [33]. According to [34], the basic assumptions that must be met for the outputs of ANOVA to be valid include independence of errors, linearity in continuous variables, absence of multicollinearity, and lack of strongly influential outliers. Additionally, there should be an adequate number of events per independent variable (covariate) to avoid an overfit model, with commonly recommended minimum “rules of thumb” ranging from 10 to 20 events per covariate. As Bayesian ANOVA was performed, testing one variable against each factor at a time, the sample size used in the present study fulfils all the assumptions. The same authors would even state that sometimes this requirement cannot be met, for instance, when the individuals in the sample are limited, which is common to all donkey breeds [1,2]. 

## 3. Results

Table 1 presents the outputs of the analysis of Bayesian inference of Pearson’s correlations between environmental levels of selenium (μg/L) and levels of Se (μg/L) and Vit E (mg/L) in and between Miranda jennies and foals.

A statistically significant correlation (*p* < 0.05) was found between soil levels of Se and Vit E for both foal and jennies, and the levels of Se was only found in foals and not in jennies. Location had a significant effect on blood Se levels in jennies. Jennies had significantly higher levels of Se in Atenor (42.412 μg/L), where soil Se levels were higher (0.97 mg/Kg) (*p* < 0.05) (Table 1 and Appendix A).

A significant effect (*p* < 0.05) for foal sex was found, i.e., Se levels were 17 times higher in female foals in this subsample of the population. Significant differences (*p* < 0.05) for Vit E (mg/L) blood levels were found with female foals presenting around twice the value than that of male foals (5.347 and 2.425 mg/L, respectively) (Appendix A). No significant influence of BCS was found for Vit E level or Se for both jennies and foals (Appendix A) (*p* > 0.05). The age of the jenny was found to be linearly related to Vit E (Pearson’s correlation coefficient: −0.24) and Se (Pearson’s correlation coefficient: 0.174 ) levels (*p* > 0.05), but age did not affect levels of either element in foal (*p* > 0.05) (Pearson’s correlation coefficient: Vit E 0.346 and Se 0.304) (refer to Appendix A).

Significant differences were found for the Se mean value in jenny (*p* < 0.01) and foal pairs (*p* < 0.05) and in Vit E levels (*p* < 0.01) in foals when we compared possibilities of diet, specifically forage fed to the donkeys (Appendix A). More specifically, the levels of Se in jennies and foals, and of Vit E in foals, statistically differed (*p* < 0.001) when either hay was provided as a supplementation and depending on whether an oat base (straw or grain) was provided or not. The supply of hay in the donkeys’ diet seemed to condition Se levels in both jennies and their foals, in a ratio of one time higher in jennies and 0.1 times lower in foals. Similarly, Vit E levels in foals were also 0.2 times lower, while its levels in jennies remained unaffected. 

In regards to supplements provisions (Appendix A), no significant difference was found for Vit E for either foals or jennies (*p* > 0.05). Extra feed supplements consisted of three-fourth of the total amount fed by weight as it has often been suggested to be recommendable [35]. However, significant differences for Se levels (*p* < 0.01) were found across diets: 1 = no supplement; salt ad libitum; and diet 3 = extra feed supplied. Se levels were slightly higher when extra feed was provided compared to when salt blocks were offered ad libitum to jennies (0.1 times higher) and moderately higher in their foals after birth (0.5 times higher). Contrastingly, when no supplementation was provided, Se levels were almost five times lower in jennies and almost six times lower in foals. 

The occurrence of foal death was statistically significantly (*p* < 0.05) conditioned by Vit E levels in jennies. Foaling mean Vit E levels ranging from 3.585 to 5.307 mg/L were associated with the occurrence of foal weakness. Contrastingly, higher levels that doubled the values reported to cause death or mummification seemed to be linked to the occurrence of foals born with carpal flexural deformities (mean = 11.520 mg/L). On the other side, the levels of Se in jennies did not significantly affect foal survival or the incidence of other problems (Appendix A). 

In the jennies, there were no significant differences (*p* < 0.05) between Vit E (mg/L) or Se (μg/L) concentrations between prefoaling and postfoaling blood levels (*F* (1,31) = 0.393 and *p* = 0.535 and *F* (1,31) = 0.637 and *p* = 0.431, respectively). However, when the previous foal had been born dead, Se levels in jennies were found to be significantly higher (63.273 μg/L as opposed to 27.206 μg/L, when the foal had been born alive). Se levels were higher in first parturition, but reduced to half when two previous foaling have occurred. These levels decreased by 30% when the number of previous parturitions was eight.

No significant differences (*p* > 0.05) were found across the different months during which sampling took place for either Vit E or Se in jennies or their foals. However, when breeding took place in March, April, May, July, and October, there was a tendency to find higher Se levels in jennies along with higher Se and Vit E levels in foals (Appendix A). A parabolic trend was described with an increasing trend for the first stage of the cycle with the higher values being reported for March, April, and May, in which the peak was reached. This parallelly occurring higher levels of blood Se in foals and jennies suggests a continuous supply of Se that may occur while the foal nurses and which may slightly cause the levels of Vit E to parallelly increase. A parabolic trend was seen in Vit E levels with jennies with the peak being reached in May but differences across months of breeding were only slight. After May, a decrease in Se was seen in jennies and foals, and a decrease in Vit E levels for foals was observed in October. No differences occurred when foaling months were compared (*p* > 0.05). Table 2 reports a summary of monthly bred and foaled jennies and the number of the dead foals.

The 17% of the jennies had previously foaled an alive foal in one additional occasion, 33% had previously foaled alive foals in two additional occasions, while the 8% of the jennies had previously foaled alive foals 8 times with an average number of foals born per jenny of 2.15.

Significant differences (*p* < 0.05) for Vit E and Se for jennies (not for foals) may occur across owners. These differences may be attributed to factors comprised within the owner factor other than location, forage, or availability of supplements (which were tested in this study, Appendix A). However, a detailed analysis of these potential factors may be necessary to identify them and determine their effects.

## 4. Discussion

For healthy donkeys, Bazzano et al. [6] reported that foals (regardless of their gender) had significantly lower Vit E concentrations than jennies, as the mean of Vit E in foals was 5.92 µmol/L, while in jennies, Vit E levels increased up to 7.81 µmol/L. The same authors reported mean Se levels in foals and jennies to be 0.05 and 0.11 µg/mL, respectively, but the values were not significantly different. These results agree with the findings of our study, for both Vit E and Se, which suggested that Se levels may be lower in foals than active adults. Se levels in donkeys may be affected by gender with jacks having higher levels compared to jennies. Bazzano et al. [6] reported similar findings with the mean Se levels in females (0.05 µg/mL) to be half the values reported for males (0.12 µg/mL). As it can be observed in Appendix A, similar findings were reported by our study with higher Se levels in jack foals when compared to jennies. 

No significant difference (*p* > 0.05) for Se and Vit E levels either in jennies or foals was observed in the posterior means across body score condition categories (Appendix A). To the knowledge of the authors, no relationship has been suggested in literature, up to this date, between body score condition and Se or Vit E levels. The lack of relationship may be supported by the data of Bazzano et al. [6], and no previous study has reported the relationship between Se and Vit E levels and lipid metabolism in donkeys. 

When diet elements were evaluated (Appendix A), levels for Se in foals were significantly lower (18.850 μg/L = 0.01885 ppm) when oat (straw with grain) and hay had been provided to the animals than when oat was provided exclusively or when other grain diets had been provided. However, Se levels in jennies fed the same diet was significantly higher (59.820 μg/L = 0.059820 ppm) when compared to the other feeding options. Significant results (*p* < 0.05 at least) were reported when hay was additionally supplied based on reports in literature. For instance, Mayland [36] demonstrated that an increase in Se levels may be related to the availability of plants, depending on a series of factors including soil pH, the oxidation–reduction potential and mineral composition of the soil, rate of artificial fertilization, and rainfall. Contextually, hay has been reported to be a relatively poor source for Vit E. This supports the slightly significantly (*p* < 0.05) higher levels for Vit E found in jennies and foals when fed oats. The two latter options showed considerably higher results, when compared to the situations in which a different diet was provided.

Contextually, selenomethionine (SeMet) is the major selenocompound in cereal grains, grassland legumes, and soybean [37]. For example, proportion of Se as SeMet in corn, rice, wheat, and soybeans reaches levels of 45.5–82%, 54.9–86.5%, 50.4–81.4%, and 62.9–71.8%, respectively [38]. Some grains, such as corn, oats, or barley, have been reported to naturally contain 20–30 IU of Vit E per kilogram of dry matter, although they may lose some of their Vit E during storage [39], which may support the higher significant levels (*p* < 0.05 at least), when oat is supplied compared to other dietary possibilities. 

Our findings suggest that different kinds of dietary supplementation may significantly condition the levels of Se; however, non-significant results were observed for Vit E. In these regards, to the knowledge of the authors, although interaction between Se levels and NaCl has not been reported in literature in animals’ nutrition studies before, studies in plant seed supplementation have indicated Se levels somehow buffer the antioxidant defense and reduces hypersaline (NaCl rich) environments, which may support our findings, as higher Se levels were found when salt bloc were provided ad libitum [40]. Still, even if supplier reported a 99.9% purity, as Se levels in salt blocks was not tested, the possibility of Se contamination cannot be discarded.

This interaction of NaCl has also been reported for other compounds such as verbacoside [41]. The effect of verbacoside on 1,2-dimyristoyl-snglycero-3-phosphoglycerol membranes (DMPG) was still noticeable at physiological ionic strengths (100–150 mM NaCl)**.** However, this effect was only abolished at very high salt concentrations (400–500 mM NaCl [41]. Contextually, dietary supplementation with verbacoside significantly decreased milk levels of saturated fatty acids and increased monounsaturated fatty acids, and vitamins A and E (*p* < 0.01) levels. Parallelly, when mares’ serum parameters were evaluated after verbacoside supplementation, the total cholesterol, triglycerides, bilirubin, ALT, and TBARs significantly decreased, while Vit E significantly increased [42]. In the aforementioned study, Vit E levels increased depending on whether diet was supplemented (7.39 μmol/L) or not (4.91 μmol/L). Casamassima et al. [43] suggested the lack of the effect of Vit E on the blood lipid profile in sheep and lambs whose diets had been supplemented with verbacoside reporting no significant changes in blood total lipids even if these were fed on diet containing controlled levels of Vit E and Se. This may suggest, in the light of our results, that lipid metabolism mechanisms linked to Se and Vit E levels may differ across species, with supplementation to regulate Vit E levels being more effective in donkeys.

Recommended daily intake of Se for donkeys has been suggested to be ~2 mg/day or in the range of 0.1–0.15 mg/100 Kg BW [44]. Karren et al. [17] suggested that depending on the diet supplied to mares, the level of Se could change. These authors provided four types of diet (pasture, pasture + grain mix, pasture + grain mix + Se, or pasture + Se). Mares fed diets based on pasture and pasture + Se received, approximately, 100% of the National Research Council’s (NRC) Nutrient Requirements. However, jennies fed diets of pasture + grain mix and pasture + grain mix + Se received 120% of suggested requirements. Therefore, foal plasma and muscle Se concentrations were greater when dams were fed the supplemental grain mix (*p* = 0.04 and 0.02, respectively) and supplemental Se (*p* < 0.001). The outcomes from this study support our results (Appendix A), and they suggested that the maternal plane of nutrition and Se supplementation may affect mare and foal plasma, muscle, and colostrum Se concentrations, but not glutathione peroxidase activity. 

Effects of owner management or practices affect the overall profiles as well. Aganga et al. [45] reported that when donkeys are working regularly but did not have adequate grazing time, owners had to provide supplemental feed. Appendix A evidences the high relative importance of feeding among other owner handling tasks. Upjohn et al. [46] suggested the importance of how the economic situation of a country where the donkeys live may influence the quality of the food provided by owners, which could ultimately affect health and development of donkey. Therefore, the influence of owners’ management practices associated with feeding can greatly influence the wellbeing of their donkeys. Our study suggests that Se levels and Vit E concentrations may widely vary with respect to the diet provided. Other studies have looked at the importance of Vit E supplementation close to foaling and how supplementing with liquid forms of alpha tocopherol may have improved absorption and bioavailability for foals [24,47]

Our study demonstrated that the age factor in jennies may significantly affect Vit E and Se levels, which was in line with the results of other authors (Appendix A). Vit E levels may reduce as age progresses, while a contrary trend is described for Se concentrations, which tend to increase. However, the rate of decrease in Vit E concentrations is about twice the rate of increase in Se concentrations. Contrastingly, foals reported both increasing levels of Vit E and Se as age progresses, with single and double times higher rates for Se concentrations and Vit E levels, respectively, when compared to that of jennies. This may suggest the high repercussion of Se concentrations in the dam on the levels for both elements in their foals. 

In these regards, Bazzano et al. [6] reported that the mean of Se in adult nonlactating jennies (4–13 years) and adult pregnant nonlactating jennies (5–12 years) were 0.05 µg/mL and 0.11 µg/mL, respectively. Furthermore, the values for Vit E in the two groups mentioned above were 7.81 µmol/L (pregnant adults not lactating) and 8.98 µmol/L, respectively. According to our results, levels of the parameters studied (Se and Vit E) in donkey foals are lower during first stages. This statement may have been supported by Smith and Burden [48] as these authors suggested that age may condition the access to food resources, which may explain why younger newborn foals have lower parameters for both nutrients. Similarly, Bazzano et al. [6] reported similar levels, 0.05 and 0.08 µg/mL, respectively, of Se for suckling foals (1–4 months of age) and weanlings/yearling (6–24 months of age). Likewise, Vit E levels for the same two groups afore described were 5.92 and 10.26 µmol/L, respectively, which may support our results.

As reported in Appendix A, posterior mean levels ranging from 3.585 to 5.307 mg/L were associated with the occurrence of foal weakness but independent, with the presence of a mummified twin fetus in the uterus associated to an endometritis or with foals found dead after non-assisted parturition. In contrast, notably higher levels, which doubled the values reported to cause death or mummification, seemed to be linked to the occurrence of foals born with carpal flexural deformity (mean of 11.520 mg/L). Carpal flexural deformity has been associated with Se and Vit E deficiencies [16,49]. According to literature, mares fed diets deficient in Se may result in foals developing WMD but carpal flexure deformity has yet to be reported [18,50]. Our results may suggest that Se deficiency in jennies may cause early foal death, and muscular dystrophy may be related to an overdose or supplementation of Se occurring at some point during gestation. However, there was no difference found in jennies for Se concentrations pre- and post-foaling.

Bazzano et al. [51] showed that nutrient requirements of jennies greatly increased in the last trimester of pregnancy. Similarly, Smith and Burden [48] suggested that the demands of the growing fetus only exceed the normal requirements in the final 3 months of pregnancy in pregnant jennies. So that, digestible energy allowances should be increased by 11% above maintenance in the 9th month, 13% in the penultimate month, and 20% in the final month of pregnancy. 

Foals’ Vit E (mg/L) levels after foaling significantly (*p* < 0.05) doubled when compared to the levels of their dams when they were in foal. This finding may suggest that apart from blood supply during gestation, a higher Vit E intake may be obtained by the foals by ingesting milk. Jennies giving birth during the autumn–winter period yielded more milk than donkeys foaling in spring–summer period since seasonal thermal stress can have a detrimental effect on the quantity and quality of milk. This may support our results as a greater milk supply may translate to a greater supply of these elements. Slight interstudy differences may be based on the seasonal climatic differences between the areas in which the different studies took place. In these regards, Cosentino et al. [52] affirmed that the effect of foaling season and lactation stage may condition milk production, which may be higher in summer at 30 days and 60 days (1.58 and 1.78 L, respectively) and in spring at 120 days (1.25 L). 

Moreover, no significant differences were found for either Vit E or Se in jennies or their foals across the different months during which the study took place, however, when the breeding took place in March, April, May, July, and October, higher Se levels were found in jennies, while the foals had higher Se and Vit E levels. A parabolic trend was described with an increasing trend for the first stage of the cycle with the higher values being reported for March, April, and May (in which the peak was reached). According to Reilly [53], higher levels of Se were found in milk during spring and summer season when compared to autumn to winter, which may support our results, and suggest that increased levels may derive from an increased obtention of Se from feeding sources in dams, which may be transferred to their foals.

This parallelly occurred for Se levels in foals and jennies, which suggest a continuous supply of Se that may occur while the foal nurses and which may slightly cause the levels of Vit E to parallelly increase. A possible explanation for the change in levels of Se and Vit E across months and seasons is likely due to the change in forages and bioavailability of these elements per location. 

## 5. Conclusions

Se and Vit E levels are lower in foals (higher in male than female foals) than active adults jennies, with Se and Vit E levels being lower during first stages of postnatal development of foals. Neither the month of sampling nor the period influences pre- or postpartum sampling. However, interlocation and owner differences suggest that environmental levels and husbandry practices may contribute to donkeys being Se and, consequently, Vit E deficient or the contrary. Among owner handling practices, feed management (hay supply) and supplement provision directly impacts on Se levels in jennies and foals and indirectly conditions Vit E levels in foals, although jennies Vit E levels are not affected, which may be indicative of the importance of Se transfer during jenny/foal interaction during nursing. 

## Figures and Tables

**Table 1 animals-11-01772-t001:** Bayesian inference for Pearson’s linear correlations between environmental levels of selenium (μg/L) and levels of selenium (μg/L) and vitamin E (mg/L) in and between Miranda jennies and foals with independence of the place where sampling took place.

Parameters	Variables	Jenny (*n* = 12)		Foal (*n* = 10)	
		Vitamin E (mg/L)	Selenium (μg/L)	Vitamin E (mg/L)	Selenium (μg/L)
Environment	Soil selenium (mg/kg)	0.092	0.598	−0.133	0.306
		(−0.239)–(0.403) *	(0.328)–(0.776)	(−0.439)–(0.199) *	(−0.022)–(0.576) *
Jenny (*n* = 12)	Vitamin E (mg/L)	–	0.002	−0.166	−0.088
			(−0.321)–(0.327) *	(−0.475)–(0.155) *	(−0.409)–(0.234) *
	Selenium (μg/L)	0.002	–	0.147	0.213
		(−0.321)–(0.327) *		(−0.175)–(0.459) *	(−0.102)–(0.518) *
Foal (*n* = 10)	Vitamin E (mg/L)	−0.166	0.147	–	0.413
		(−0.475)–(0.155) *	(−0.175)–(0.459) *		(0.133)–(0.673)
	Selenium (μg/L)	−0.088	0.213	0.413	–
		(−0.409)–(0.234) *	(−0.102)–(0.518) *	(0.133)–(0.673)	

* Significant effect (*p* < 0.05).

**Table 2 animals-11-01772-t002:** Summary of monthly bred and foaled jennies and the number of the dead foals.

	Jennies Bred	Jennies Foaled	Dead Foals
January			
February			
March	1		
April	7	3	1
May	2	4	1
June		1	
July	1		
August		1	
September			
October	1		
November		1	
December			

## Data Availability

Data will be made available from corresponding author on reasonable request.

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
