# Peer review of "Selenium and Vitamin E Concentrations in Miranda Jennies and Foals (Equus asinus) in Northeast Portugal"

_animals, 2021, doi:10.3390/ani11061772_

Round 1

Reviewer 1 Report

None

Author Response

Response: We thank the reviewer for his/her work on this manuscript.

Reviewer 2 Report

In my opinion, the Manuscript has improved in quality and is becoming more understandable. Undoubtedly the work of the reviewers and especially the work done by the authors has contributed enormously to this. However, I do not agree with some of the answers issued by the authors, and with some of the modifications made.

I still think that the revised article lacks sufficient entity to be published as an extended communication. In my opinion it should be considered as Short Communication. I do not agree with the authors when they indicate that requesting by the reviewers to expand some aspects of the methodology, the results and the discussion would lead to a much more extensive manuscript. My recommendation to publish as Shor Comunicaction well given by the entity of the contributed work, not by its extension.

And of course, for this, some aspects that are superfluous should be eliminated, and the length of the Manuscript should be shortened.

I still think that the number of animals sampled, even with the statistical study carried out, is too small to establish reference values, and above all to draw clear conclusions.

Capital letters have been grossly abused in some names. For example Selenium (L-69, 71, 245, etc), also Vitamin E (L-69), Soil (L-189), Lichens (L249). Review all the text.

Material and methods.

The methodology about the collection and processing of the samples has greatly improved. But, I don't see how and why lichen sampling is done. I understand that it is a bioindicator of environmental quality, but at no time is it justified in the Introduction, or in the methodology used.

I do not understand the difference between the groups with different feeding: L-151 “The donkeys had daily access to pasture, supplemental forage, grain (e.g. oats) and salt blocks depending on the owner’s animal feeding habits”. As far as I know, the salt blocks only contribute Cl and Na, but they cannot contribute selenium or vitamin E at all. Therefore, I do not understand to take into account this differentiation. And much less than that is considered in the statistical study.

The parameters are measured in soil and in lichens, but they are not done in administered food. However, it is done in the Discussion. Isn't it a major methodological error?

Results.

The tables of the previous version have been revised and corrected. Now they are easier to understand.

Tables have also been abused in supplementary material. The statistical study has only been “copied” when the important thing is to describe the findings of that statistical study.

Discussion

Although the Discussion is more complete than in the previous version, I think that it is still quite ambiguous, and should focus more on those aspects that have been found in the results, such as differences based on sex, owner, diet, etc., and the no differences depending on the sampling month, between before and after delivery, etc. On the other hand, aspects such as the levels in the food, or the predisposition of the foals to suffer some diseases as a consequence of these levels, which have not been measured in the experiment, have been discussed.

Conclusions

They have been modified, but I still think that they are not truly conclusions drawn from this research. The first part of the conclusions (L481 and 482) are correct. The rest do not.

Lines 483 and following should be brought to Discussion. The authors give their opinion on why the levels of Se and Vit E can be modified in jennies and foals, … but that should not be the conclusions of an investigation, but the clearly verified findings. I would recommend that the authors include in this section the existence of differences based on sex, the owner's management (mainly feeding), the location, and on the contrary that neither the month of sampling nor the period influences pre or postpartum sampling.

It is important that these conclusions are summarized in 1 or 2 sentences.

References

Without exhaustively reviewing the references used, there are a couple of them that stand out to me:

L-557; 9. Kaneko, J.; Harvey, J.; Bruss, M. Clinical Biochemistry of Domestic Animals. J. Vet. Res. Commun 1997, 22, 293-294.

Without a doubt that reference is incorrect. It is a widely used book, and not a J. Vet magazine article. Res. Commun. In addition, there is a more updated version of this book, which I recommend to the authors:

Kaneko, J.J.; Harvey, J.; Bruss, M.; eds. Clinical Biochemistry of Domestic Animals, 5th Ed; Academic Press: San Diego, USA, 1997; ISBN 978008052919.

Kaneko, J.J.; Harvey, W.J.; Bruss, M.L. eds. Clinical Biochemistry of Domestic Animals; 6th ed.; Elsevier- Academic Press: San Diego, USA, 2008; ISBN 978-0-12-370491-7.

L-562; 12. In, N. Nutrient requirements of horses. National Academies Press: Washington, DC, 2007. 562

I don't know exactly what this reference is. Review and modify.

Author Response

Reviewer 2

Review Report Form

Comments and Suggestions for Authors

In my opinion, the Manuscript has improved in quality and is becoming more understandable. Undoubtedly the work of the reviewers and especially the work done by the authors has contributed enormously to this. However, I do not agree with some of the answers issued by the authors, and with some of the modifications made.

Response: We thank the reviewer for his/her kind comments, and for his attention to our manuscript. We understand that he/she disagrees with some of our answers, and will try to address new comments to reach an agreement.

I still think that the revised article lacks sufficient entity to be published as an extended communication. In my opinion it should be considered as Short Communication. I do not agree with the authors when they indicate that requesting by the reviewers to expand some aspects of the methodology, the results and the discussion would lead to a much more extensive manuscript. My recommendation to publish as Shor Comunicaction well given by the entity of the contributed work, not by its extension.

Response: Still, we disagree. We attach a list of papers in which the same or very similar topic (or even a rather limited approach) was developed, two of them during the last 5 years. Furthermore, the methods in the paper open new opportunities for data processing for those who are not familiarized with Bayesian approaches to solve frequentist experiences. As a result, we leave this at the journal office discretion. It is not the extension, but the paper working with an understudied species, hence whose results have a direct application and transference to the donkey medicine sector, and the fact that the paper uses an alternative approach, among other things, which provide the paper with sufficient entity as to be considered an extended communication.

  1. Shawaf, T.; Almathen, F.; Meligy, A.; El-Deeb, W.; Al-Bulushi, S. Biochemical analysis of some serum trace elements in donkeys and horses in Eastern region of Kingdom of Saudi Arabia. Veterinary world 2017, 10, 1269-1274, doi:10.14202/vetworld.2017.1269-1274.
  2. Bazzano, M.; McLean, A.; Tesei, B.; Gallina, E.; Laus, F. Selenium and Vitamin E Concentrations in a Healthy Donkey Population in Central Italy. J Equine Vet Sci 2019, 78, 112-116, doi:10.1016/j.jevs.2019.04.003.
  3. Ludvikova, E.; Jahn, P.; Pavlata, L.; Vyskočil, M. Selenium and vitamin E status correlated with myopathies of horses reared in farms in the Czech Republic. Acta Veterinaria Brno 2005, 74, 377-384.
  4. Al-Busadah, K. Trace-elements status in camels, cattle and sheep in Saudi Arabia. Pakistan Journal of Biological Sciences (Pakistan) 2003.

And of course, for this, some aspects that are superfluous should be eliminated, and the length of the Manuscript should be shortened.

Response: According to the journals guide for authors, Articles should have a main text of around 3000 words at minimum and should have more than 30 references. If this is the minimum for articles, one could expect short communications may extend shorter or at least equally. Our paper is almost 10000 words long. Hence, of course even if we reduce the superfluous details that the reviewer suggests, we understand that these are not going to constitute more than half of what is written, what reinforces our thought that the paper must be considered as a full article. The paper was reduced to around 8500 words.

still think that the number of animals sampled, even with the statistical study carried out, is too small to establish reference values, and above all to draw clear conclusions.

Response: We agree with the reviewer on the fact that sample is limited. However, according to literature and provided our results, our sample may be in the lower limit to issue valid results, which theoretically supports on the information already presented in the discussion and introduction of the paper. In these regards, credible intervals are provided. In Bayesian statistics, a credible interval is an interval within which an unobserved parameter value falls with a particular probability. It is an interval in the domain of a posterior probability distribution or a predictive distribution. The standard tool for estimating the credibility of a statistical result is the confidence interval (CI), which describes the precision of an estimate. Hence the results could be considered to be cross validated.

Capital letters have been grossly abused in some names. For example Selenium (L-69, 71, 245, etc), also Vitamin E (L-69), Soil (L-189), Lichens (L249). Review all the text.

 Response: Suggestion was followed, both in body text and supplementary material.

Material and methods.

The methodology about the collection and processing of the samples has greatly improved. But, I don't see how and why lichen sampling is done. I understand that it is a bioindicator of environmental quality, but at no time is it justified in the Introduction, or in the methodology used.

Response: As we clarified this information was described in previous papers, anyway, as we find the reviewer is not sure about its inclusion, we decided to remove it from the paper.

I do not understand the difference between the groups with different feeding: L-151 “The donkeys had daily access to pasture, supplemental forage, grain (e.g. oats) and salt blocks depending on the owner’s animal feeding habits”. As far as I know, the salt blocks only contribute Cl and Na, but they cannot contribute selenium or vitamin E at all. Therefore, I do not understand to take into account this differentiation. And much less than that is considered in the statistical study.

Response: We do not understand the author inquiry. Interaction between micronutrients has often been reported in literature. Indeed, that is what these results may suggest, possibly rather than salt blocs contributing with higher Se levels, there is an interaction between selenium and NaCl which may lead to a greater absorption of selenium as suggested when salt blocs are made available ad libitum. This interaction has been reported for plants, for instance in:

 Hasanuzzaman, M., Hossain, M.A. & Fujita, M. Selenium-Induced Up-Regulation of the Antioxidant Defense and Methylglyoxal Detoxification System Reduces Salinity-Induced Damage in Rapeseed Seedlings. Biol Trace Elem Res 143, 1704–1721 (2011). https://doi.org/10.1007/s12011-011-8958-4

The parameters are measured in soil and in lichens, but they are not done in administered food. However, it is done in the Discussion. Isn't it a major methodological error?

Response: it is not a major methodological error as we designed our study and chose the statistical tests to perform according to the information that we had. If we had measured the levels of selenium in food, these would have been measured as a quantitative trait. Consequently, ANOVA directly on the data could not have been performed, but other analyses such as correlations, among others. As that information was not available, we decided to consider it a categorical variable and study whether the particular event of feeding different possibilities or diet influenced or not Sel and Vit E in jennies and foals. What the reviewer suggests as a major methodological error would be other study but not ours. How can something that we did not do because it was not an aim in ours study, nor analysed, nor discussed be an error? Indeed, we think that measuring levels in diet (with the complexity that diet involves) may give way to a rather complex and different study. What we did in discussion, as the only information that we know is how the presence in diet of the different dietary combinations (Oat (straw with grain), Oat (straw with grain) and hay and others) influences Jenny selenium (μg/L), Foal vitamin E (mg/L), Foal selenium (μg/L) when hay is provided. Our discussion focused on providing evidences and suggesting potential reasons for this based in literature, but we never discussed our results in regards selenium levels in our food because we did not measure them. We revised discussion to make sure that any potential confusion on us discussing diet nutritional composition is removed.

Results.

The tables of the previous version have been revised and corrected. Now they are easier to understand.

Response: Thank you.

Tables have also been abused in supplementary material. The statistical study has only been “copied” when the important thing is to describe the findings of that statistical study.

 Response: We removed Table S9 and S11 as we understand they may not add much to the aims in the present study and as the results that they report have already been added somehow in the body text. The rest of tables is the minimum necessary to support results, discussion and conclusions. We added justification for each of them in the discussion.

Discussion

Although the Discussion is more complete than in the previous version, I think that it is still quite ambiguous, and should focus more on those aspects that have been found in the results, such as differences based on sex, owner, diet, etc., and the no differences depending on the sampling month, between before and after delivery, etc. On the other hand, aspects such as the levels in the food, or the predisposition of the foals to suffer some diseases as a consequence of these levels, which have not been measured in the experiment, have been discussed.

Response: We reinforced discussion in the sections in which the reviewer requested that we should. Furthermore, we clarified how discussion sections supported or contradicted our results and linked each specific part to the specific table presenting the results for readers to be able to easily find which the statistical basis for our comments.

Conclusions

They have been modified, but I still think that they are not truly conclusions drawn from this research. The first part of the conclusions (L481 and 482) are correct. The rest do not.

Lines 483 and following should be brought to Discussion. The authors give their opinion on why the levels of Se and Vit E can be modified in jennies and foals, … but that should not be the conclusions of an investigation, but the clearly verified findings. I would recommend that the authors include in this section the existence of differences based on sex, the owner's management (mainly feeding), the location, and on the contrary that neither the month of sampling nor the period influences pre or postpartum sampling.

Response: Conclusions were rewritten leaving lines 481 and 482. 

It is important that these conclusions are summarized in 1 or 2 sentences.

Response: We reduced the conclusion section as much as possible.

References

Without exhaustively reviewing the references used, there are a couple of them that stand out to me:

L-557; 9. Kaneko, J.; Harvey, J.; Bruss, M. Clinical Biochemistry of Domestic Animals. J. Vet. Res. Commun 1997, 22, 293-294.

Without a doubt that reference is incorrect. It is a widely used book, and not a J. Vet magazine article. Res. Commun. In addition, there is a more updated version of this book, which I recommend to the authors:

Kaneko, J.J.; Harvey, J.; Bruss, M.; eds. Clinical Biochemistry of Domestic Animals, 5th Ed; Academic Press: San Diego, USA, 1997; ISBN 978008052919.

Kaneko, J.J.; Harvey, W.J.; Bruss, M.L. eds. Clinical Biochemistry of Domestic Animals; 6th ed.; Elsevier- Academic Press: San Diego, USA, 2008; ISBN 978-0-12-370491-7.

Response: We corrected it.

L-562; 12. In, N. Nutrient requirements of horses. National Academies Press: Washington, DC, 2007. 562

I don't know exactly what this reference is. Review and modify.

Response: We modified it.

Reviewer 3 Report

Please, find enclosed all comments and suggestions.

Thank you

Author Response

Reviewer 3

Please, remove as it is not appropriate according to the data presented.

Response: We removed it, as suggested.

Specify... nutritional values, biochemical values?

Response: It was clarified.

Interesting but not necessary. Can be removed.

Response: It was removed as suggested.

Too long. Needs to be summarized so the whole introduction will fit in one page.

Response: It was summarized to fit one page.

Editing issue I believe. Vitamin E is not the abbreviation of glutathione peroxidase.

Response: We agree and corrected.

repeated at the end of the sentence.... may be removed

Response: Removed.

Remove hypotheses.

Response: Hypotheses were removed.

Is it enough animals to call it a survey?

Response: We removed the term survey from title.

is it possible to be extravenous?

Response: No, it is intravenous.

remove figure

Response: Figure was removed.

This table can be removed and data displayed within the text.

Response: This was a suggestion of a previous reviewer.

Is the N sufficient to come to those conclusions? These are very variable parameters and the number of individuals needed to come to conclusions need to be large.

Response: We understand the reviewer concern. However, this is the reason why we decided to use a Bayesian approach and given post cross validation through the analysis of credibility intervals had resulted appropriate, conclusions can be issued validly.

very descriptive but many P values are missing

Response: P-values were provided.

It would be more useful to talk about Spring, summer, fall and winter. It would make more sense to people living out of Portugal.

Response: We decided to use months as all seasons were not represented due to the characteristics of sampling. However, pretty similar results can be presumed given seasons cluster months in groups of three.

Is it possible that the supplement had Se contamination?

Response: It was not possible. This was pointed out by another reviewer.

this percentage is relative to what?

Response: We clarified it is proportion of Se as SeMet

Again, please check data for accuracy.

Response: As mg/L and µg/mL use equivalent units in numerator and denominator and are ratios they report the same values and are interchangeable.

Round 2

Reviewer 3 Report

Dear authors,

Most of the suggestions and comments were addressed and questions answered. Three comments were added to the latest version of the manuscript. They aim to get further precision in some aspects.

Thank you

Author Response

Reviewer 3 2nd round

I suggest to remove it because it is redundant. If animals are not slaughtered, blood can only be collected intravenously.

Response: We removed it.

My point here is not the unit of measure itself (which is interchangeable with mg/L as the authors well stated in their response) but the fact that authors [5] reported 34.57 mg/L and the present study reports 0.05 ug/ml. It is big difference. How do the authors explain that?

Response: We revised the data and manuscripts cited, and we presume that Shawaf made a mistake in reporting the units which indeed should be µg/L. As the only reason that these authors provide is differences in levels being ascribed to diet sources, hence we feel this info must be removed.

The authors answered to the previous questions suggesting that the supplement could not be contaminated with selenium. Was this supplement analyzed for Se? How they can be sure about it?

The label of the supplier read composition white salt block was NaCl with a 99.9% guaranteed analysis. Hence, it can be presumed that even if it was contaminated, levels would be negligible as to be causative of the effect detected.

This manuscript is a resubmission of an earlier submission. The following is a list of the peer review reports and author responses from that submission.

Round 1

Reviewer 1 Report

The authors measured Se and Vit E plasma levels from Miranda jennies pre and post-foaling  and plasma levels in their foals to make a comparison between them and other external or animal health- and history-related factors. This work adds information on the hematochemical parameters of this species, for which horse ranges have been used for a long time and which, as shown in different manuscripts, are not always comparable with the donkey ranges.

Title: Pertinent.

Summary:  Following the rules of the review, summary must represent a brief resume of the work where clear statement of the problem addressed, aims and objectives, results, conclusion from the study and its relative social impact are presented.  In your summary results and conclusions are missing.

Abstract: The study background (presented only in the summary) and conclusions are not present in the abstract. In line 34 you wrote “serum” instead of “plasma”. Lines 41-42 should be rewritten in a better form. Moreover, you did not mention the fact that you calculated the correlation between the Se amount in the soil and the Se levels in the animals. When rewriting the summary and abstract, please consider that they must be 200 words long.

Keywords: “Foal, Jennies, Donkey health, Pregnancy, Lactation” could be added.

Introduction: It is fine overall, but there are some typographical errors in lines 58, 181, 260, 269, 295.

The starting hypothesis is not clearly stated.

Lines 55-58: In this form, the sentence is not clear and should be rewritten.

Line 59: Ref. 4 is more pertinent if placed in line 63.

Line 66: Since in ref. 5 the authors measured only minerals and not vitamins, it would be better to add other references.

Line 76: it could be more logical to move the sentence “Selenium-vitamin E injections can be used for short-term supplementation [9,10]” after ref. [11]. Reference [9] is too old and you should look for more recent works on the topic.

Line 86: Refs. 14 and 15 are about calves and not foals.

Line 96: Ref. 22 is not pertinent, since the authors measured Se levels and thyroid hormones. In the introduction of this work you will find references on the neurological and immunitary implications of Se and vit E deficiency. When mentioning foals and pathologies, you should specify which animal species (horses or donkey) your references are referring to.

Line 100: Ref 27 is old, it would be better to add a more recent one.

Materials e methods: Was the sampling in the jennies performed before feeding? Was the sampling in the last pregnancy collected at the same time for all the dams? Here, you do not mention the fact that the animals were housed in two different places and you do not specify how they were distributed. Moreover, for every single month, you should specify how many jennies breeded and foaled, the number of the healthy foals, how many jennies foaled before (specifying if the foal was dead or alive) and the number of parturitions for single jenny. This information could be inserted in a new table.  The method for obtaining the Se levels in the soil should be mentioned in this section and not only in Figure 1 (if you used data from a reference, please cite it)
I have more observations about it: you did your best with the few available animals.  Your study has been performed from spring to autumn, and this is a long period where the climate changes a lot. Indeed, the Se concentration in the soil mostly depends from rainfall events, 
because of the leaching action  of rainwater (see “Selenium deficiency risk predicted to increase under future climate change” Jones et al., 2017): for the Se levels in the soil you used data from a 15-year old publication (Ventura et al., 2005) and you neither  got  them directly nor considered or referred to the weather of that specific year (2018). In Ventura et al., were the surveys about large areas of the country or did they give specific values for Paradela e Atenor? Were they the same areas where donkeys lived? Indeed, concentration values in the soil may change from different areas.  It would be nice if you could send me the work from Ventura et al, 2005.

Which soil and lichen Se concentration did you use to calculate the correlation in Tab. 1? Did food and water used for the animals come from the same zone they were housed in? These details should be added and all the limits of your study should be discussed. The BCS partitioning for obtaining groups used in the ANOVA is not clearly specified in this section. The last part of the statistical analysis, where you justify the use of the Bayesian method in order to overcome the small size of the animal sample, is redundant and can be summarized leaving more space for the other information.

Line 132: which is the percentage of the extra feeding respect to the intended food ration?

Line 144-147: since here you are presenting laboratory analysis, it would be better to move this part in the next paragraph.

Line 162: for the seek of consistency, you should add the reference number. Moreover, since the reference is the same, you can avoid to insert it also in line 158.

Results:

Line 236: A significant correlation for Vit E is present in both animals, while for Se this correlation is observed in foals only. Moreover, it P<0.05 instead of P>0.05.

Table 1.: The review requires title and caption to be inserted. Heading in the columns are missing and, in particular, you do not specify Se in the environment variables. Columns and rows should be separated with lines for ease of consultation. It would be better to indicate the number of jennies and foals. Moreover, it should be observed that the table not only describes the correlation between the Se levels in the soil and the levels of Vit E and Se in jennies and foals, but it also gives information about the mutual correlation between Vit E and Se within and between the two groups (dams and foals). This concept should be better indicated even in the statistical analysis, in the results and in the caption. You should specify in the table caption which coefficient you are referring to.

Fig. 1: Figure title is missing. If the figure has been taken from Ventura et al., 2005 and has been subsequently modified, this should be indicated (mod.). The idea is captivating, but you should use a sharper picture of the donkeys.

Supplementary Tables: Here, title and caption must be inserted and the abbreviations should be explained. TableS.1: If they are all jennies, why do you present values for males? Referring to Tab. S4: it could be interesting to insert a plot of Bayesian inference for Pearson’s linear correlation (Age vs Vit. E in jennies). The calculated coefficient should be indicated in the table caption. Tab.S7: Does the sentence “Does not apply” mean “healthy foal”? In the table, did you use only post-foaling sample results for jennies? Same question for TabS.10. Why did you use the breeding month and not the sample month?  Please, clarify these points and add the information in the captions or in the main text.

Line 256: “the jenny” appears twice.

Lines 258-276: Please insert the Se and Vit E values you are referring to.

Line 284-286: In which table do you present these values? Where are the pre-foaling e post-foaling results?

Lines 288-290: It should be indicated if the differences were significant or not.

Lines 292-293: Where are the sampling data for every single month during which the sample took place?

Discussion: It is logically structured, but I have some observations:

Lines 311-317: You should be more precise indicating that in Bazzano et al, 2013 no extremely significant difference has been found for Vit E and Se between foals and jennies, but only between foals and jacks in the Se levels.

Lines 319-322: You inverted female and male Se values when citating Refs 5 and 13, where Se-male > Se-female, and these differences were not very relevant after all. Moreover, you did not collect sample from jacks, so these sentences should be rewritten.

Line 340: The percentages you show refer to the Se percentage in SeMet form respect to the total Se amount contained in the food. This has to be specified, otherwise these percentages are meaningless.

Line 348-354: Ref 44 is about horses and not donkeys.

Line 364: 0.1-0.15 mg/100 Kg BW.

Line 364-471: Ref 17 is about horses and not donkeys.

Line 397: The Vit. E value in adult nonlactating jennies is 7.72 μmol/L and in adult pregnant non lactating jennies is 7.81 μmol/L.

Line 402-405: You should be more precise indicating that in Bazzano et al, 2013 no extremely significant difference has been found for Vit E and Se between foals of different ages.

Line 412: Ref. 52 is about WMD, not Carpal flexural deformity and it is not pertinent.

Line 413: Ref 18 is about horses and not donkeys.

Lines 427-430:  It is a repetition and can be eliminated.

Line 448: Citation number is missing in the bibliographic entry.

After the changes requested, this document deserves to be published for its future impact in the field of equine internal medicine.

Author Response

Reviewer 1

Comments and Suggestions for Authors

The authors measured Se and Vit E plasma levels from Miranda jennies pre and post-foaling  and plasma levels in their foals to make a comparison between them and other external or animal health- and history-related factors. This work adds information on the hematochemical parameters of this species, for which horse ranges have been used for a long time and which, as shown in different manuscripts, are not always comparable with the donkey ranges.

Title: Pertinent.

Summary:  Following the rules of the review, summary must represent a brief resume of the work where clear statement of the problem addressed, aims and objectives, results, conclusion from the study and its relative social impact are presented.  In your summary results and conclusions are missing.

Response: Thank you. The new statement has been added as requested.

Line 29-32: Our study suggests that Vit E and Se levels can have a major impact and effect on foal health and mortality levels. Multiple factors including location, diet, management practices, parity and time of breeding and age of the jenny may affect blood profiles in jennies which ultimately may affect the profiles of her foals. 

Abstract: The study background (presented only in the summary) and conclusions are not present in the abstract.

Response: They are now present.

Study background: An increase in donkeys being treated by practitioners in recent years has led to an increase interest in finding more information on basic reference values for donkeys.

Conclusion: Diets, location, parity, age may all affect blood profiles of jennies and ultimately foal health.

In line 34 you wrote “serum” instead of “plasma”.

Response: Line 34 Serum has been changed to plasma

 Lines 41-42 should be rewritten in a better form.

Response:

Line 41-42, now Lines 42-43 read:

Low Vit E levels (5.307 mg/L) in jennies were related to foal mortality.

The abstract has been re-written.

Moreover, you did not mention the fact that you calculated the correlation between the Se amount in the soil and the Se levels in the animals.

Response: We clarified it in the abstract.

When rewriting the summary and abstract, please consider that they must be 200 words long.

Response: Simple Summary is now 186 words long

Abstract is now 199 words long

Keywords: “Foal, Jennies, Donkey health, Pregnancy, Lactation” could be added.

Response: Thank you! We have added: Foals; Jennies, Donkey health, Pregnancy, Lactation

Introduction: It is fine overall, but there are some typographical errors in lines 58, 181, 260, 269, 295.

Response: Line 58- now reads: Today, donkeys are being kept and managed more intensively and carefully than ever before [1].

Line 181 (now Line 222-6): Due to the difficulties of working in loco with an endangered breed with a small population scattered through different [2,3],, the small sample size and sample distribution violated parametric assumptions (Shapiro-Francia P<0.05, Levene’s test P<0.05), therefore, Bayesian inference Pearson correlation function was used to characterize the posterior distribution of the linear correlation between environmental (soil and lichens Se levels), jennies and their foals’ concentrations of Se (μg/L) and Vit E (mg/L).

Line 260 (now 303-305.): Significant differences were found for the Se mean value in jenny and foal pairs and in Vit E levels in foals. When we compared possibilities of diet, specifically forage fed to the donkeys, differences were recorded (Supplementary Table 5).

Line 269 (now line 315): salt blocs now changed to salt blocks

Line 295 (now line 338-39): However, when breeding took place in March, April and May, there was a tendency to find higher Se levels in jennies along with higher Se and Vit E levels in foals (Supplementary Table 10).

The starting hypothesis is not clearly stated.

Response: Line 151: The hypothesis was to test the relationship of Se and Vit E levels in jennies compared to foals and jennies with higher levels of Vit E and Se prior to foaling with produce foals with compromising health conditions such as limb flexure deformities and have improved foal survival rates.

Lines 55-58: In this form, the sentence is not clear and should be rewritten.

Response: Line 95-96 now reads: Donkeys have traditionally been used as draft animals but growing interests in using donkeys in different capacities such as production animals for milk, meat and skin along with an increased value related to donkeys being kept as performance and companion animals has led to an increase interest on how to improve the care of donkeys.  Great efforts to preserve endangered breeds of donkeys such as the Miranda has led to more scientific literature and specie specific information on donkey health becoming available, [4].

Line 59: Ref. 4 is more pertinent if placed in line 63.

Response: Ref 4 has been added to this line 63 (now 104).

Line 66: Since in ref. 5 the authors measured only minerals and not vitamins, it would be better to add other references.

Response: A reference dealing with Vit E and minerals was added.

Line 76: it could be more logical to move the sentence “Selenium-vitamin E injections can be used for short-term supplementation [9,10]” after ref. [11]. Reference [9] is too old and you should look for more recent works on the topic.

Response: Reference was chamged to Muirhead, T.L.; Wichtel, J.J.; Stryhn, H.; McClure, J.T. The selenium and vitamin E status of horses in Prince Edward Island. Can Vet J 2010, 51, 979-985. Dating to 2010.

Line 86: Refs. 14 and 15 are about calves and not foals.

Response: Now line 131: Thank you the correction has been made, foals removed and animals (e.g calves) added.

Line 96: Ref. 22 is not pertinent, since the authors measured Se levels and thyroid hormones. In the introduction of this work you will find references on the neurological and immunitary implications of Se and vit E deficiency. When mentioning foals and pathologies, you should specify which animal species (horses or donkey) your references are referring to.

Response:  reference was removed.

Line 100: Ref 27 is old, it would be better to add a more recent one.

Response: Line 635 new reference has been added from a leading equine expert in Vit E research and up to date, 2015, Finno, C.J.; Estell, K.E.;Katzman, S.; Winfield, L.; Rendahl, A.; Textor, J.; Bannasch, D.L.;Puschner, B.  Blood and cerebrospinal fluid a-tocopherol and selenium concentrations in neonatal foals with neuroaxonal dystrophy.   J. Vet. Inter. Med. 2015, 29, 1667-75.

Materials e methods: 

Was the sampling in the jennies performed before feeding?

Response: All animals ate before sampling. Eventualy, it would not have been humanely reasonable having animal welfare in consideration, to restrict food intake when jennies were recovering from parturition and milking.

Was the sampling in the last pregnancy collected at the same time for all the dams?

Response: Parturitions occured during night or dawn on different days, hence sampling was perfomed accordingly.

Here, you do not mention the fact that the animals were housed in two different places and you do not specify how they were distributed.

Response: Added.

Moreover, for every single month, you should specify how many jennies breeded and foaled, the number of the healthy foals, how many jennies foaled before (specifying if the foal was dead or alive) and the number of parturitions for single jenny. This information could be inserted in a new table. 

Response: We added the information requested in Table 1 and in the three lines below as requested.

The method for obtaining the Se levels in the soil should be mentioned in this section and not only in Figure 1 (if you used data from a reference, please cite it) 

Response: Added.

I have more observations about it: you did your best with the few available animals.  Your study has been performed from spring to autumn, and this is a long period where the climate changes a lot.

Indeed, the Se concentration in the soil mostly depends from rainfall events,  
because of the leaching action  of rainwater (see “Selenium deficiency risk predicted to increase under future climate change” Jones et al., 2017): for the Se levels in the soil you used data from a 15-year old publication (Ventura et al., 2005) and you neither  got  them directly nor considered or referred to the weather of that specific year (2018).

Response: On average, the Mediterranean climate registers temperatures of around 20 °C throughout the year, a temperature that increases markedly in summer and decreases a few degrees below 10 °C in winter. In general terms, it is a stable and pleasant climatic zone. For instance, as suggested by Couto, et al. [5], the climate characteristic of the region is Mediterranean. Mean annual rainfall is around 554.7 mm, and rainfall is persistent until the end of May, leading to a high pasture availability in the beginning of June. Mean monthly temperature recorded is around 15 °C in June and 23 °C in July. Hence and considering the relative proximity between the two locations this may have not presumable drastically affected levels in the area.

In Ventura et al., were the surveys about large areas of the country or did they give specific values for Paradela e Atenor? Were they the same areas where donkeys lived? Indeed, concentration values in the soil may change from different areas.  It would be nice if you could send me the work from Ventura et al, 2005.

Response: Selenium Levels in Mainland Portugal | SpringerLink

Which soil and lichen Se concentration did you use to calculate the correlation in Tab. 1?

Response: Data was obtained from the European Soil Data Centre (ESDAC) and was compared to that reported in Ventura (2005).

Did food and water used for the animals come from the same zone they were housed in?

Response: Yes. This was of course considered provided levels in food may depend on the levels of selenium present in their origin location.

These details should be added and all the limits of your study should be discussed.

The BCS partitioning for obtaining groups used in the ANOVA is not clearly specified in this section. The last part of the statistical analysis, where you justify the use of the Bayesian method in order to overcome the small size of the animal sample, is redundant and can be summarized leaving more space for the other information.

Response: Added. Partitioning for BCS was added as well. Bayesian sample size suitability fragment was summarized.

Line 132: which is the percentage of the extra feeding respect to the intended food ration?

Response: Extra feed supplements consisted of ¾ of the total amount fed by weight as it has often been suggested to be recommendable.

Line 144-147: since here you are presenting laboratory analysis, it would be better to move this part in the next paragraph.

Response: This information has been deleted since it’s replicated under lab analysis (line 214-17).

Line 162: for the seek of consistency, you should add the reference number. Moreover, since the reference is the same, you can avoid to insert it also in line 158.

Response: Change has been made and reference number [13] has now been added and reference last name deleted in both line 162 (now line 219) and line 158 now line 222.

Results:

Line 236: A significant correlation for Vit E is present in both animals, while for Se this correlation is observed in foals only. Moreover, it P<0.05 instead of P>0.05.

Response: Now line 301-3. A statistically significant correlation (P<0.05) was found between soil levels of Se and Vit E for both foal and jennies, and the levels of Se was only found in foals and not in jennies. 

Table 1.: The review requires title and caption to be inserted. Heading in the columns are missing and, in particular, you do not specify Se in the environment variables. Columns and rows should be separated with lines for ease of consultation. It would be better to indicate the number of jennies and foals. Moreover, it should be observed that the table not only describes the correlation between the Se levels in the soil and the levels of Vit E and Se in jennies and foals, but it also gives information about the mutual correlation between Vit E and Se within and between the two groups (dams and foals). This concept should be better indicated even in the statistical analysis, in the results and in the caption. You should specify in the table caption which coefficient you are referring to.

Response: Caption was inserted. Heading in columns is present as well. The format of the journal states not to add lines between rows. Number of jennies and foals was indicated. Changes requested were applied.

Fig. 1: Figure title is missing. If the figure has been taken from Ventura et al., 2005 and has been subsequently modified, this should be indicated (mod.). The idea is captivating, but you should use a sharper picture of the donkeys.

Response: Figure was modified and changes were applied as requested.

Supplementary Tables: Here, title and caption must be inserted and the abbreviations should be explained. TableS.1: If they are all jennies, why do you present values for males?

Response: There were male foals as well.

Referring to Tab. S4: it could be interesting to insert a plot of Bayesian inference for Pearson’s linear correlation (Age vs Vit. E in jennies).

Response: Bayesian inference for Pearson correlation methodologically is based in Bayesian linear regression, so does Bayesian ANOVA. We decided not to include a plot provided Pearson correlations are monotonic hence the information of the plot and that of the table would be redundant.

The calculated coefficient should be indicated in the table caption. Tab.S7: Does the sentence “Does not apply” mean “healthy foal”? In the table, did you use only post-foaling sample results for jennies? Same question for TabS.10. Why did you use the breeding month and not the sample month?  Please, clarify these points and add the information in the captions or in the main text.

Response: No significant differences were found across the different months during which sampling took place for either Vit E or Se in jennies or their foals as suggested in the body test. Hence no Table is presented for this factor.

Line 256: “the jenny” appears twice.

Now line 324 and the second “jenny” has been removed, now reads: The age of the jenny was found to be linearly related to Vit E and Se levels, but age did not affect its foal’s levels of either element (Supplementary Table 4).

Lines 258-276: Please insert the Se and Vit E values you are referring to.

Line 258 (now Line 324) reads: The age of the jenny was found to be linearly related to Vit E (-0.24 mg/L) and Se (l0.174 mg/L) levels, but age did not affect its foal’s levels of either element, (Vit E 0.34.6 mg/L and Se 0.34 μg/L) (Supplementary Table 4).

Line 284-286: In which table do you present these values? Where are the pre-foaling e post-foaling results?

Response: Now lines 353-354. These values were not included in Tables but reported in the body text to reduce the number of tables enclosed.

Lines 288-290: It should be indicated if the differences were significant or not.

Response: Now lines 355-357

Lines 292-293: Where are the sampling data for every single month during which the sample took place?

Response: Table 1 was added comprising this information.

Discussion: It is logically structured, but I have some observations:

Lines 311-317: You should be more precise indicating that in Bazzano et al, 2013 no extremely significant difference has been found for Vit E and Se between foals and jennies, but only between foals and jacks in the Se levels.

Response: Now lines 393-4: Now reads: The same authors reported lower levels of Se in foals 0.05 ug/mL compared to 0.11 ug/mL in jennies yet the difference was not significant.

Lines 319-322: You inverted female and male Se values when citating Refs 5 and 13, where Se-male > Se-female, and these differences were not very relevant after all. Moreover, you did not collect sample from jacks, so these sentences should be rewritten.

Response: Now lines 396-403 reads: Some studies have shown that Se levels in donkeys may be affected by gender with jacks having higher levels compared to jennies. Both Shawaf, et al., [5] and Bazzano et al., [13] reported higher levels in jacks.

Shawaf, et al. [6] reported Se levels for jacks (34.57±1.71 mg/L, μ±SD) than for jennies (30.09±2.96 mg/L, μ±SD). Bazzano, et al. [7] reported similar findings with the mean Se levels in females (0.05µg/mL) to be half the values reported for males (0.12 µg/mL). We found similar findings with higher Se levels in jack foals when compared to jennies.

Line 340: The percentages you show refer to the Se percentage in SeMet form respect to the total Se amount contained in the food. This has to be specified, otherwise these percentages are meaningless.

Response: Now line 421-22-Now reads: For example, total SeMet in corn, rice, wheat and soybeans reaches levels of 45.5-82%, 54.9-86.5%, 50.4-81.4% and 62.9-71.8%, respectively [8].

Line 348-354: Ref 44 is about horses and not donkeys.

Response: Now line 432- jennies has been replaced with mares

Line 364: 0.1-0.15 mg/100 Kg BW.

Response: Now line 452- “g” has been added to Kg

Line 364-471: Ref 17 is about horses and not donkeys.

Response: Now line 453-4- jennies has been changed to mares

Line 397: The Vit. E value in adult nonlactating jennies is 7.72 μmol/L and in adult pregnant non lactating jennies is 7.81 μmol/L.

Response: This has been added

Line 402-405: You should be more precise indicating that in Bazzano et al, 2013 no extremely significant difference has been found for Vit E and Se between foals of different ages.

Response: Now line 490-91: The word similar has been added to the sentence to suggest the difference was not large.

Line 412: Ref. 52 is about WMD, not Carpal flexural deformity and it is not pertinent.

Response: These sentences have now been changed.

Line 413: Ref 18 is about horses and not donkeys.

Response: Now line 500 jennies changed to mares

Lines 427-430:  It is a repetition and can be eliminated.

Response: This sentence has now been deleted.

Line 448: Citation number is missing in the bibliographic entry.

Response: Reference was added.

After the changes requested, this document deserves to be published for its future impact in the field of equine internal medicine.

Response: Thank you so much for the extremely kind and supportive comment! We have tried our best to include all of your very helpful edits and comments to improve the paper.

Reviewer 2 Report

The revised article lacks sufficient entity to be published as a full communication. In my opinion it should be considered as Short Communication.

The Material and Methods section should be rewritten. Much importance is given to the statistical study, but other aspects such as sample collection, sampling locations, processing of those samples, and other factors that will be taken into account in the subsequent statistical study are very weak. I would like all the factors mentioned in the Results to have been described in Material and Methods.

For example: a study of Vit levels is carried out. E and selenium in lichens, but at no time is it indicated how they have been sampled, how it has been measured, and the first allusion to this assessment is made in L-185 (in statistical study) and in L-238 (in Results).

The number of animals sampled is very low. Foals may be difficult to sample, but female donkeys are not. And they are too few jennies. I doubt the statistical studies are conclusive, even with Bayesian Inference. I remind you that in some group there are 2-3 animals.

Table 1 is not understood. A line appears to be missing. I would recommend that the authors redo this table, and make it simpler. I would ask for a table showing the values found, with descriptive statistics and the number of animals in each group.

Results. Tables have been misused in Supplementary Material. In my opinion they do not contribute much and less in that format, in which the statistical study carried out has been "copied". Review and simplify these tables.

Discussion: in my opinion it is the most correct part. A comparison is made with other References, although it is difficult to follow it, since we do not know the values obtained by the authors. However, there is no discussion of the statistical study carried out, nor of the inferences found.

The conclusions are very ambiguous, and very long. And they do not reflect what was found in the investigation. I would recommend rewriting them and doing 1-2 sentences.

Author Response

Reviewer 2

The revised article lacks sufficient entity to be published as a full communication. In my opinion it should be considered as Short Communication.

Response: The present paper deals with an underdealt topic in an understudied species using a not commonly applied statistical method to process the data. The description of the methodology used was reduced up to a point in which the loss of information still permitted the understanding of the methods applied in the present research. Reducing more information as to fulfil the reviewer suggestion would imply a reduction or mistreatment of all the data collected and evaluated in the present research. As a result, reducing the paper would then make future studies not able to repeat the current studies even if additional donkeys were added. Furthermore, even if statistical section was reduced reviewer suggests to include information rather than reducing which is totally comprehensible, but which makes it complex o make it fit the shape of a short communication.

The Material and Methods section should be rewritten. Much importance is given to the statistical study, but other aspects such as sample collection, sampling locations, processing of those samples, and other factors that will be taken into account in the subsequent statistical study are very weak.

Response: Thank you for your suggestion. The materials and method section has now been re-written.

I would like all the factors mentioned in the Results to have been described in Material and Methods.

For example: a study of Vit levels is carried out. E and selenium in lichens, but at no time is it indicated how they have been sampled, how it has been measured, and the first allusion to this assessment is made in L-185 (in statistical study) and in L-238 (in Results).

Response: Information was added as suggested.

The number of animals sampled is very low. Foals may be difficult to sample, but female donkeys are not. And they are too few jennies. I doubt the statistical studies are conclusive, even with Bayesian Inference. I remind you that in some group there are 2-3 animals.

Response: we understand and have addressed the sample size is low and that’s partially due to sampling an endangered breed of donkey, meaning the numbers are not high. Also, one of the aims of the study was to sample pregnant jennies so we could correlate values of Vit E and Se in the jenny and then in her foal. Sampling additional females that were not pregnant or in foal would not contribute to the correlation of Vit E and Se from the dam to the offspring but thank you for the suggestion. Furthermore, the relatively narrow confidence intervals presented in supplementary material account for the validity of findings and of the conclusions drawn after such findings.

Table 1 is not understood. A line appears to be missing. I would recommend that the authors redo this table, and make it simpler. I would ask for a table showing the values found, with descriptive statistics and the number of animals in each group.

Response: Table was corrected.

Results. Tables have been misused in Supplementary Material. In my opinion they do not contribute much and less in that format, in which the statistical study carried out has been "copied". Review and simplify these tables.

Response: Tables were presented in the format used in other already used papers presenting similar statistical outputs.

Discussion: in my opinion it is the most correct part. A comparison is made with other References, although it is difficult to follow it, since we do not know the values obtained by the authors. However, there is no discussion of the statistical study carried out, nor of the inferences found.

Response: The complete statistical explanation, which was reduced as suggested by this and other reviewers, discussed the reasons of the use of the statistical methodologies described in the present paper. For instance, this section was developed aiming to issue a comparable method to be able to relate the findings in the present research to other findings present in literature (For instance the term of significance is a frequentist one, not a Bayesian one, but a method to extrapolate results was applied). To our knowledge, Bayesian inference has not been used for studies of this kind and in our vision, it enables opening other opportunities to process data that can be helpful in certain contexts in which sample limitations threat the validity of conclusions.

The conclusions are very ambiguous, and very long. And they do not reflect what was found in the investigation. I would recommend rewriting them and doing 1-2 sentences.

Response: Thank you for your suggestion. The conclusion has been reviewed and modifications have been made.

Reviewer 3 Report

Survey of Selenium and Vitamin E Concentrations in Miranda Jennies and Foals in Northeast Portugal

  • In general, the simple summary seemed to ramble and not having particular purpose. I would recommend major revisions to address need to understand Se and Vit E in Miranda donkeys and the signf
  • Line 19- refrain from use possession (donkeys’)
  • Line 23- define at first use (Se and Vit E)
  • Line 24- refrain from use of possession (foals’)
  • Line 25- previous to meaning gestation or pregnancy or in-utero during fetal development?
  • Line 26- what about gestation?
  • Line 37- suggestion to reword throughout to peripartum and postpartum versus pre and post-foaling
  • Line 38-define “higher” or add value with SE and/or statistical significance
  • Line 41-reword sentence for clarity
  • Line 44- two fold vs. double
  • Line 48-64 -May need to define and contrast the genus of Equus – Equus caballus vs Equus asinus
  • Line 48-writing does not flow, seems blocked in writing style from first to second paragraph; does not flow well for easy reading
  • Line 50- can they vary in color (brown to black)?; long-haired; remove “very friendly”; what about population size and/or lifespan?
  • Line 52- further describe event or significance of “Baudette Poitou of France” for the novice reader
  • Line 55- Extra space between (. and Recently)?; eliminate “to now”
  • Line 56- Define “production”
  • Line 58 (.) vs. (,)
  • Line 59- define “carefully”
  • Line 60- refrain from use of “there’s” and use “there is”; this statement conflicts with Line 57-58 comment
  • Line 66- “there is”
  • Line 67- “and”; define “Vit E” at first use; “an”
  • Line 80- “Se” to “Selenium”; introduction seems to bounce around, recommend discussing Se and Vit E separate and then together.
  • Line 83- age as in foal vs. mature or and/or stage of production, parity, etc.
  • Line 79-93 could be made one paragraph with first discussing placental function, lactation physiology, and neonate requirements and growth
  • Line 81- “neonate” vs newborn
  • Line 82- maybe just say stage/phase of production
  • Line 83- are levels species and/or breed specific. What is the best way to determine Vit E and Se status and accuracy?
  • Line 86- “adequate” vs. appropriate and define requirements for gestating dams
  • Line 97-do not start sentence with abbreviation; “and result in milk entering the lungs and causing aspiration pneumonia”.
  • Line 105- “sex” not “gender”
  • Line 106- Is Portugal typically deficient in Se in soil and forages?
  • Line 110- “Our second objective was to determine the relationship”; While the objectives are clearly given there needs to be additionally information provided about the hypothesis, at least from the stand point of the soil or regions that may be of particular importance in Portugal where the Miranda donkeys are prevalent.
  • Line 115- Why is no information given about “soil” sampling and analyses in Materials and Methods? Why was forage (hat and pasture not analyzed?)
  • Line 120- “removed or excluded” vs. “discarded”
  • Line 121- Significance of year?; “Body condition score (BCS) was recorded”
  • Line 122- “Samples were collected from Miranda jennies (n=12) and foals (n=10). There were two foals that were not sampled as both were born dead, one prematurely at 10.5 m of gestation and the other full term.”
  • Line 126- Is the Pearson and Ouassat BCS scale one that is appropriate to use for both mature donkeys and foals? Furthermore, foal morphometrics of crown-to-rump, hearthgirth, and body weight, and vigor at birth would be a better measurement for the young foals.
  • Line 128- What type of pasture? What type of forage? So all of the donkey jennies were fed different diets and housed in the different pastures?
  • Line 134- “parity” vs. “number of foals foaled”; “foal sex” vs. “foal gender”
  • Line 135- “weak” vs. “week”; consider use of “vigor” vs. “difficulty standing”; did all get colostrum?
  • Line 139- Why was colostrum not sampled if the foals were allowed to nurse?
  • Line 142- “or”? Where not all foals sampled?
  • Line 145- use correct symbol for degrees
  • Line 147- include citation
  • Line 155-Duplicated statement from Line 142
  • Line 158 & 162- Consistency in citations [x] vs. (Bazzano et al., 2019)
  • Line 180 – “loco?”
  • Line 181- remove (,)
  • Line 182- italicize P
  • Line 196- “sex” not “gender”. “BCS” as already abbreviated
  • Line 215- What is the power of analysis needed? Preform calculation and include.
  • Line 236- italicize P
  • Table 1- difficult to read table. Major improvement is needed. Include actual P values.
  • Figure 1- The donkey image is very pixelated
  • Line 251- italicize P; “sex” not “gender”; not a population rather a subsample of a population.
  • Line 254 & 324 &329 - “BCS” as already abbreviated
  • Line 328- A brief literature search indicates that there has been reports of BCS and selenium status and/or supplementation (doi: 3168/jds.2017-14196 )
  • Citations- Provide doi for those available. “A” vs. “a” in animals in 1 &2 and throughout. Questionable use of citations such as 43. Effects of Antioxidants in The Horse. Is this peer reviewed article?

Author Response

Reviewer 3

Survey of Selenium and Vitamin E Concentrations in Miranda Jennies and Foals in Northeast Portugal

  • In general, the simple summary seemed to ramble and not having particular purpose. I would recommend major revisions to address need to understand Se and Vit E in Miranda donkeys and the signficance

Response: Thank you for your suggestion. We have revisited the summary and stressed the importance for for understanding Vit E and Se values in relationship to preserving the Miranda Donkey.

  • Line 19- refrain from use possession (donkeys’)

Response: This has been changed and now reads: Despite the importance of donkeys through history and their productive resuscitation during the last decades, reference values for common elements are not yet readily available.

  • Line 23- define at first use (Se and Vit E)

Response: Line 23 now reads: The aims of this study are to determine baseline Se and Vit E concentrations for Miranda donkeys both jennies and foals.

  • Line 24- refrain from use of possession (foals’)

Response: Foals’ has been removed and sentence now reads: Miranda donkeys are considered to be endangered and its possible that Se and Vit E may be associated with foal survival.

  • Line 25- previous to meaning gestation or pregnancy or in-utero during fetal development?

Response: Line now reads: Critical points may be identified related to overdosing or deficient levels of Se and Vit E, at different stages of development of gestation in utero during fetal development, parturition and post foaling

  • Line 26- what about gestation?

Response: This has been defined: in utero during fetal development

  • Line 37- suggestion to reword throughout to peripartum and postpartum versus pre and post-foaling

Response: Line 37 now reads: The aims of this study were to measure Se and Vit E levels in plasma from Miranda jennies peripartum and postpartum  and in their foals to compare blood profiles of the jenny and foal pair related to overall foal health.

  • Line 38-define “higher” or add value with SE and/or statistical significance

Response: Values have now been added for soil and jennies to show statistical significance

  • Line 41-reword sentence for clarity

Response: Line 41 now reads: Foals with a mean Vit E of 3.585 to 5.307 mg/L showed signs of weakness yet carpal flexural deformities were observed when the average was 11.520 mg/L.

  • Line 44- two fold vs. double

Response: This sentence has been changed and two-fold/double has been excluded.

  • Line 48-64 -May need to define and contrast the genus of Equus – Equus caballus vs Equus asinus

Response: Line 48-64 is now 245 and Equus asinus has been added after donkeys.

  • Line 48-writing does not flow, seems blocked in writing style from first to second paragraph; does not flow well for easy reading

Response: Thank you for the suggestion. We have gone back and reviewed the first and second paragraph and tried to improve the follow per your helpful suggestion. The first paragraph of the introduction now reads:

There are close to 200 breeds of donkeys (Equus asinus) and most are considered endangered (178/200) [2,3]. The Miranda donkey is considered to be tall, black to brown in color with long hair, and a docile temperament. This breed is found in northeast Portugal. Traditionally, the Miranda donkey was used for agrarian purposes such as farming potatoes and helping harvest grapes for wine production. The breed helped save the Baudette Poitou of France. Recent efforts have examined risks factors that may be associated with the high foal mortality rates that are being seen by Miranda donkey owners. Great efforts to preserve endangered breeds of donkeys such as the Miranda has led to more scientific literature and specie specific information on donkey health becoming available along with interest in donkeys being used and valued as production and performance animals, [4]. Today, we find donkeys being kept and managed more intensively and then ever before [1]. Unfortunately, there’s still a lack of information related to the basic understanding of donkey health parameters [4]. As a result, even routine diagnostic tests such as blood chemistry and haematological test parameters may be based off of lab standards set for horses which are not always the same for donkeys. This complicates routine internal medicine and general health care for donkeys.

  •  
  • Line 50- can they vary in color (brown to black)?; long-haired; remove “very friendly”; what about population size and/or lifespan?

Response: Line 50 now line 246 reads: The Miranda donkey is considered to be tall, black to brown in color with long hair, and a docile temperament.

  • Line 52- further describe event or significance of “Baudette Poitou of France” for the novice reader

Response: Line 52 now line 249 reads: The breed helped save the Baudette Poitou, a French breed of donkey that is similar in terms of phenotype and temperament by outcrossing the remaining population with the Miranda donkey.

  • Line 55- Extra space between (. and Recently)?; eliminate “to now”

Response: Thank you for the suggestion but this sentence has been removed.

  • Line 56- Define “production”

Response: Thank you for the suggestion but this sentence has been removed.

  • Line 58 (.) vs. (,)

Response: The “,” has been removed.

  • Line 59- define “carefully”

Response: “carefully has been removed and sentence restructured.

  • Line 60- refrain from use of “there’s” and use “there is”; this statement conflicts with Line 57-58 comment

Response: Thank you this has been corrected to “there is”

  • Line 66- “there is”

Response: Thank you this has been corrected to “there is”

  • Line 67- “and”; define “Vit E” at first use; “an”

Response: Thank you this has been corrected to read, Vitamin E (Vit E)

  • Line 80- “Se” to “Selenium”; introduction seems to bounce around, recommend discussing Se and Vit E separate and then together.

Response: Now line 244-248 and reads Vit E and Se are essential nutrients for equine [9], and the relationship of their levels with foal-dam prepartum and postpartum has often been addressed in literature. One study showed that levels of Vit E and Se can vary in donkeys according to age and sex with foals having lower levels of Vit E compared to adults [7]. Defencies in these nutrients may compromise the health of donkeys, especially a newborn foal, growing donkey, or lactating jenny.

  • Line 83- age as in foal vs. mature or and/or stage of production, parity, etc.

Response: Now line 247 reads Deficiencies in these nutrients may compromise the health of donkeys, especially in foals, young donkeys, or lactating jenny.

  • Line 79-93 could be made one paragraph with first discussing placental function, lactation physiology, and neonate requirements and growth

Response: Thank you for the suggestion. We have worked on changing and revising the sentences into one paragraph.

  • Line 81- “neonate” vs newborn

Response: Newborn has been changed to neonate

  • Line 82- maybe just say stage/phase of production

Response: Thank you for your suggestion. This paragraph has been revised.

  • Line 83- are levels species and/or breed specific. What is the best way to determine Vit E and Se status and accuracy?

Response: The study looked at commercial dairy donkeys in Italy not a particular dairy breed and samples were taken from various groups of donkeys according to age, stage of lactation, and sex.

  • Line 86- “adequate” vs. appropriate and define requirements for gestating dams

Response: This sentence has been added: Defining appropriate or normal levels of Se and Vit E in donkeys is still being determined by sampling populations and measuring different parameters that may alter levels found when testing blood samples

  • Line 97-do not start sentence with abbreviation; “and result in milk entering the lungs and causing aspiration pneumonia”.

Response: Thank you we have updated the sentence and it now reads: White muscle disease can affect both horse [10] and donkey foals [11] and in some cases weak muscles in the throat of the foal may lead to milk entering the trachea and then the lungs causing aspiration pneumonia.

  • Line 105- “sex” not “gender”

Response: Now line 459, gender has been changed to “sex”

  • Line 106- Is Portugal typically deficient in Se in soil and forages?

Response: Yes, as reported by the phD Thesis by Catarina Isabel Alves Lourenço Galinha, which has now been cited.

  • Line 110- “Our second objective was to determine the relationship”; While the objectives are clearly given there needs to be additionally information provided about the hypothesis, at least from the stand point of the soil or regions that may be of particular importance in Portugal where the Miranda donkeys are prevalent.

Response: Thank you for your comment, we have gone back and revised the. Hypothesis and included the information in the paragraph. It now reads:

Response: The first hypothesis was to test the relationship of Se and Vit E levels in jennies compared to foals and jennies with higher levels of Vit E and Se prior to foaling will produce foals with fewer compromising health conditions such as limb flexure deformities and have improved foal survival rates. So, the first aim of the study was to determine blood profiles of Se and Vit E in jennies and their foals and compare. The second hypothesis was to test the relationship of the soil levels where jennies and foals lived and soils with higher levels of Se would likely results in foals that had a higher chance of surviving. Second objective looked at establishing a relationship between Se and Vit E levels in jennies and their foals related to the mineral composition of the soil. The third hypothesis expected reproductive history of each jenny in terms of age, number of live foals born and foals that lived past the first week would be more likely to produce healthy foals. The last objective compared the jennies’ influence on foal health/survivability by examining her parturition history, foaling record and blood profile of Se or Vit E related to their foal’s health.

  •  
  • Line 115- Why is no information given about “soil” sampling and analyses in Materials and Methods? Why was forage (hat and pasture not analyzed?)

Response: Information was extracted from an external official database upon reasonable request and contrasting results to publications as it has now been clarified.

  • Line 120- “removed or excluded” vs. “discarded”
  • Response: Please see updated paragraph below (Line 122 suggestion)
  • Line 121- Significance of year?; “Body condition score (BCS) was recorded”
  • Response: Please see updated paragraph below (Line 122 suggestion)

  • Line 122- “Samples were collected from Miranda jennies (n=12) and foals (n=10). There were two foals that were not sampled as both were born dead, one prematurely at 10.5 m of gestation and the other full term.”

Response: Section now reads (line 498-505): Blood samples were collected from 12 Miranda jennies and 10 foals. Two foals were not sampled due to premature deaths.  An intravenous 10 mL sample of blood was collected from the jugular vein of each jenny and/or foal into an EDTA (ethylenediaminetetraacetic acid) tube (VACUETTE; Greiner Bio-One GmbH, Kremsmunster, Austria). All blood samples were stored in a dark box refrigerated at 18ºC immediately after collection, transported from sampling location to laboratory. The jennies (n= 12) ranged in age from 3 to 24 years (7.9 ± 5.9 years), with an average BCS of 2.9 ± 0.59 (1 to 5 scale). The foals (n = 10, n = 6 males and n = 4 females) were sampled between day one and three after birth, 2 ± 1 days, and BCS was 2.3 ± 0.35 (1 to 5 scale).

  • Line 126- Is the Pearson and Ouassat BCS scale one that is appropriate to use for both mature donkeys and foals? Furthermore, foal morphometrics of crown-to-rump, hearthgirth, and body weight, and vigor at birth would be a better measurement for the young foals.

Response: Thank you for your suggestion on determining foal morphometrics. This is something we could include in future studies. The scale used was orginally developed from Pearson and Ouassat and now commonly used by many organizations and charities measuring BCS in donkeys (e.g. The Donkey Sanctuary) as a gold standard versus the 1-9 scale developed by Henneke which was developed for brood mares not donkeys.

  •  
  • Line 128- What type of pasture? What type of forage? So all of the donkey jennies were fed different diets and housed in the different pastures?

Response: Information on diet was gathered from an owner survey. Forages that were fed to donkeys included oat straw with grain or oat straw without grain.

  •  
  • Line 134- “parity” vs. “number of foals foaled”; “foal sex” vs. “foal gender”

Response: Now line 587: Thank you both have been changed to parity and foal sex

  • Line 135- “weak” vs. “week”; consider use of “vigor” vs. “difficulty standing”; did all get colostrum?

Response: Thank you for your suggestions both have been corrected: weak now on line 588 and vigor has replaced difficulty standing

  • Line 139- Why was colostrum not sampled if the foals were allowed to nurse?

Response: Selenium and vitamin E are involved in immune function and may influence colostrum quality when they are deficient. Dams fed a prepartum diet deficient in selenium and vitamin E produce less colostrum and lower total mass of colostral IgG than dams fed the same diet but supplemented with injections of vitamin E and selenium (Lacetera et al., 1996). However, Almeida, et al. [12] reported no significant correlations between colostrum and blood trace element levels, suggesting that the analyzed elements are subjected to a regulated transport from blood to milk, hence, we decided not to consider it.

  • Line 142- “or”? Where not all foals sampled?

Response: Yes, all foals that survived or were born a live (10/12) were sampled. Now line 498. Reads: Blood samples were collected from 12 Miranda jennies and 10 foals. Two foals were not sampled due to premature deaths.

  • Line 145- use correct symbol for degrees

Response: Now line 488, symbol has been corrected, 18° C

  • Line 147- include citation

Response: Citation has been included, reference “13” for the methodology on testing for both micro nutrients

  • Line 155-Duplicated statement from Line 142
  • Response: Now line 581-2 and reads: Once samples arrived at the lab each sample was removed from the dark box and centrifuged within 30 minutes from collection for 10 mins at 1500 g and 4°C.
  •  
  • Line 158 & 162- Consistency in citations [x] vs. (Bazzano et al., 2019)
  • Response: Thank you, the citation has been replaced with the appropriate reference #, 13.

  • Line 180 – “loco?”

Response: Meaning location? The term has been updated now line 604

  • Line 181- remove (,)
  • Response: The extra comma has been removed
  • Line 182- italicize P
  • Response: P has been italicized

  • Line 196- “sex” not “gender”. “BCS” as already abbreviated
  • Response: Now Line 620 has been updated from gender to sex and Body Condition Score has been changed to BCS

  • Line 215- What is the power of analysis needed? Preform calculation and include.

Response: You can perform hypothesis tests with Bayesian statistics. For example, you could conclude an effect is greater than zero if more than 95% of the posterior density is greater than zero. Or alternative, you could employ some form of binary decision based on Bayes factors.

Once you establish such a decision making system, it is possible to assess statistical power assuming a given data generating process and sample size. You could readily assess this in a given context using simulation.

That said, a Bayesian approach often focuses more on the credibility interval than the point estimate, and degree of belief rather than a binary decision. Using this more continuous approach to inference, you could instead assess other effects on inference of your design. In particular, you might want to assess the expected size of your credibility interval for a given data generating process and sample size. That is what we did.

  • Line 236- italicize P
  • Response: P has been italicized

  • Table 1- difficult to read table. Major improvement is needed. Include actual P values.

Response: Table was corrected considering other correlation tables present in literature. P values were added as a footnote. Indded P-values in Bayesian approaches are inferenced rather than actual P values.

  • Figure 1- The donkey image is very pixelated

  • Response: We changed. It was not pixelated. It was a mosaic trying to emulate Portuguese floors. We adapted it to follow reviewer suggestion.

  • Line 251- italicize P; “sex” not “gender”; not a population rather a subsample of a population.
  • Response: Gender has been replaced with sex (now line 721), and subsample has been added

  • Line 254 & 324 &329 - “BCS” as already abbreviated
  • Response: BCS has replaced Body Condition Score on now line 722

  • Line 328- A brief literature search indicates that there has been reports of BCS and selenium status and/or supplementation (doi: 3168/jds.2017-14196 )

Response: Thank you for the suggestion, this work is actually sited in our reference list as Karren, B.; Thorson, J.; Cavinder, C.; Hammer, C.; Coverdale, J. Effect of selenium supplementation and plane of nutrition on mares and their foals: selenium concentrations and glutathione peroxidase. J. Anim. Sci. 2010, 88, 991-997. But the study doesn’t specifically correlate BCS to Se absorption or circulating levels by saying a mare that is a BCS of 6 will have X µg/mL of Se, or a mare of a lower BCS. Thank you for the recommendation though.

  • Citations- Provide doi for those available. “A” vs. “a” in animals in 1 &2 and throughout.

Response: A in Animals has been capitalized and a “s” has been added to the end of the journal name. Thank you for the suggestion. Doi is not required by the journal style unless the article has not been assigned an issue already.

  •  
  • Questionable use of citations such as 43. Effects of Antioxidants in The Horse. Is this peer reviewed article?

Response: Thank you for your inquiry about this article, no it is not peer-reviewed but The Horse interviews experts in their field leading scientist and researchers on their topic of expertise.